# Method for MICCAI FLARE24 Challenge

Jiufu Zhu

Linyi University

**Abstract.**. In recent years, the prevalence of cancer cases has been significant, with projections indicating a substantial increase by 2024. Effective radiation therapy (RT) is crucial for treatment, yet it poses risks to surrounding organs. Accurate delineation of organs at risk (OARs) on CT images is essential to mitigate these risks. Medical imaging, particularly CT scans, plays a pivotal role in tumor diagnosis and treatment planning. Automated segmentation of tumors in pan-cancer CT scans using advanced computational techniques, such as deep learning, is pivotal for improving clinical decision-making. This paper explores a novel training strategy based on the ResUNet model to enhance segmentation accuracy. The strategy involves phased training, first without skip connections for primary segmentation and then with skip connections for detailed segmentation, aiming to improve both efficiency and precision. Experimental results demonstrate promising potential, highlighting the method's applicability and future directions in medical image segmentation.

**Keywords:** Cancer • Segmentation •

## 1    Introduction

In recent years, the number of cancer cases has been substantial. By 2024, it is projected that there will be 2,001,140 new cancer cases and 611,720 cancer-related deaths in the United States [1]. Evidence-based literature indicates that 50% of these patients will require radiation therapy (RT) [2]. However, while RT is effective in killing tumor cells, it can also damage normal tissues and lead to various complications. Therefore, it is necessary to accurately delineate the organs at risk (OARs) surrounding the tumor on CT images before formulating a treatment plan, and to minimize the radiation dose to these organs during optimization.

Medical imaging plays a critical role in the diagnosis, treatment planning, and monitoring of tumors. Computed Tomography (CT) scans provide detailed anatomical information that is crucial for identifying and characterizing tumors of different cancer types. Segmentation studies in pan-cancer CT scans refer to the process of automatically or semi-automatically delineating tumor regions in these images, regardless of cancer type, location, or patient characteristics. This approach aims to develop robust algorithms and methods capable of accurately and efficiently segmenting tumors, thus aiding clinical decision-making and improving patient care outcomes.The complexity of pan-cancer segmentation arises from the diverse morphological features of tumors, including their size, shape, density, and spatial distribution. Traditional manual segmentation methods are time-consuming, subject to

inter-observer variability, and may lack scalability for large datasets. In contrast, advanced computational techniques, especially deep learning and artificial intelligence (AI), have emerged as powerful tools for automating this process. Specifically, Convolutional Neural Networks (CNNs) models, customized for 3D

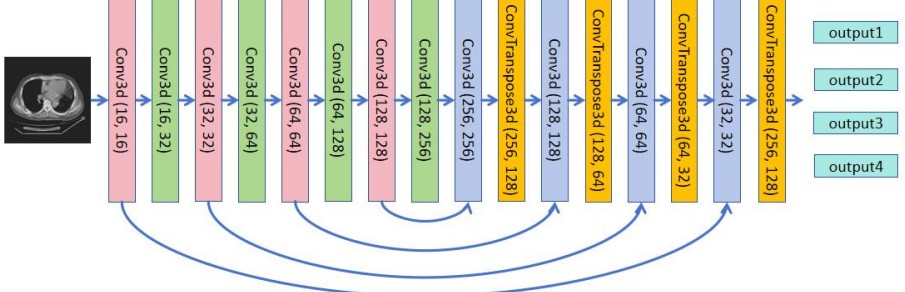

**Fig. 1.** The figure shows the architecture of the ResUNet neural network model. By incorporating residual learning, the model enhances its expressive power. The architecture consists of an encoder and a decoder, which progressively extract and restore multi-scale features from the image through a series of convolutional layers (Conv3d) and transposed convolutional layers (ConvTranspose3d).

medical image analysis, have demonstrated high accuracy and efficiency in segmenting tumors in CT scans.

In this paper, we explore the current progress in segmentation research on CT scans of all cancer types. We also assist model training by utilizing a relatively large-scale dataset. We propose a new training strategy based on the existing medical segmentation model (ResUNet) [4] by changing the training strategy. First, we complete our primary segmentation by removing skip connections and training to obtain the primary segmentation results. Based on the primary segmentation, we then add skip connections to complete the detailed segmentation, ultimately achieving the final segmentation results.

## 2    Materials and Proposed Method

### 2.1    Data & Image Pre-processing

We are using the dataset provided by MICCAI FLARE 2024 TASK 1: PAN-CANCER SEGMENTATION IN CT SCANS. It includes 5000 annotated training cases, which are partially annotated. Each case only annotates the primary lesions, with other lesions potentially unannotated (e.g., metastatic lesions). Additionally, there are 5000 unannotated training cases, of which we use only the 5000 annotated cases for training. The dataset comprises 209 cases of COVID-19 pneumonia, 1093 cases from the DeepLesion dataset for deep learning research, 488 cases from the KiTS23 dataset for kidney tumor segmentation tasks, 1010 cases from the Lung Image Database Consortium (LIDC) for lung nodule detection and segmentation, 892

cases from the Multi-modality Medical Image Segmentation Dataset (MSD), 52 cases from The Cancer Imaging Archive (TCIA) related to adrenal glands (TCIA-Adrenal), 176 cases from TCIA related to lymph nodes (TCIA-LymphNodes), 581 cases from TCIA related to non-small cell lung cancer (TCIA-NSCLC), and 500 cases of whole-body computed tomography scans (whole-bodyCT).

We will constrain the grayscale values of CT images to the range of -200 to 200 to eliminate outliers. Next, we will downsample both the CT and segmentation images to reduce data volume and resolution, resizing the dataset from 512x512 to 256x256. Additionally, we will extend slices containing organs in the segmentation images by 20 slices before and after each slice.

## 2.2    Proposed Method

**Competing Methods.** Building on the existing medical image segmentation model (ResUNet), changing the training strategy is an innovative approach that can further improve the model's performance and precision. We propose a phased training strategy: first, remove the skip connections to train for primary segmentation, yielding the main segmentation results; then, reintroduce the skip connections to train for detailed segmentation. This two-phase training strategy allows the model to initially focus on identifying primary structures and subsequently refine the segmentation at a higher resolution. This approach maintains overall segmentation accuracy while significantly enhancing the handling of details, ultimately producing more precise segmentation results. This method not only optimizes the model's training process but also enhances the robustness and accuracy of segmentation tasks in complex medical images.

# 3    Experiment

## 3.1    Environment settings

Our model is developed based on PyTorch. All models are trained from scratch. We use Dice loss to train the segmentation network. The development environment and requirements are shown in Table 1.

**Table 1.**    Development environments and requirements.

| Windows/Linux version | Ubuntu 22.04.4 LTS |
|---|---|
| GPU (number and type) | Two NVIDIA GeForce RTX 3090 24G |
| CPU | Intel(R) Xeon(R) Gold 6133 CPU @ 2.50GHz |
| RAM | 187GB |
| CUDA version | 12.2 |
| Programming language | Python 3.10.14 |
| Deep learning framework | PyTorch (torch 2.3.1, torchvision 0.18.1) |

## 3.2 evaluation measures

We adhered to the guidelines outlined by the FLARE challenge and employed the Dice Similarity Coefficient (DSC) and Normalized Surface Distance (NSD) as quantitative metrics to evaluate the segmentation results rigorously. The DSC measures the overlap between expert annotation masks and segmentation outcomes on a region-by-region basis, providing insight into how accurately the segmented regions align with ground truth annotations. Specifically, it quantifies the similarity as twice the intersection divided by the sum of the volumes of the predicted and reference segmentations. This metric is particularly valuable in assessing the fidelity of segmentation algorithms in capturing the intricate contours and boundaries of anatomical structures in medical imaging data.

$$DSC(G, S) = \frac{2|G \cap S|}{|G| + |S|}$$

# 4 Results

## 4.1 Quantitative results

We compared our method with nnU-Net's segmentation model. As shown in Table 2, our current average DSC on the FLARE 2024 validation set is 0.016. Compared to nnU-Net, which was also trained from scratch, our method demonstrates lower DSC scores on most tumor organs. This suggests that under the current experimental setup, our method requires further optimization to enhance its ability to recognize complex structures and fine features. Nevertheless, our approach significantly reduces the complexity and computational costs of model training, which is crucial for future research and applications in medical image segmentation.

**Table 2.**  DSC values

| Our | nnUnet |
|---|---|
| 0.017 | 0.331 |

The segmentation outcomes are visually depicted in Figure 2, illustrating the comparative effectiveness of our method against nnU-Net and highlighting the nuanced differences in tumor organ delineation.

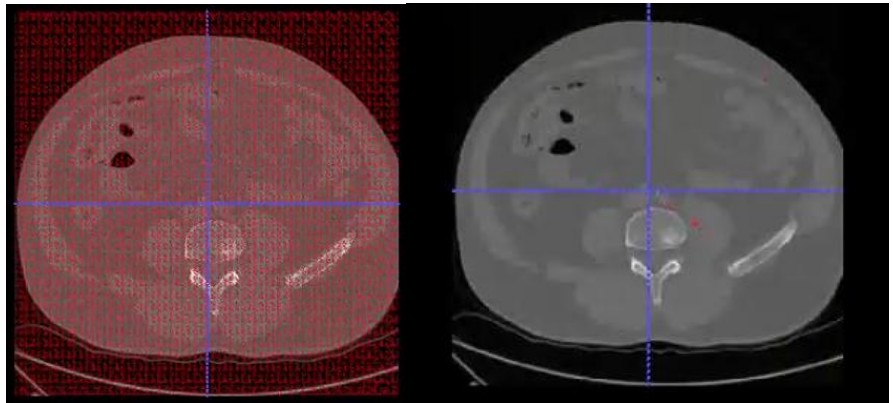

**Fig. 2.**Comparison between segmentation results of different methods.

# 5    Conclusion and Future Work

This new training strategy has demonstrated broad potential and advantages in experiments. By conducting training in stages, we can finely control the model's learning process, thereby optimizing its performance more effectively across different resolutions. Removing skip connections during the primary segmentation phase focuses on precise identification of overall structures, while reintroducing them during the detailed segmentation phase facilitates finer segmentation at higher resolutions, enhancing the model's ability to capture details and improve segmentation accuracy. This approach enables the model to handle complex structures and subtle features more effectively, opening new possibilities for medical image analysis.

Moreover, compared to traditional global training methods, our strategy significantly simplifies the model training process. Segmental training reduces computational resource demands, lowers the risk of overfitting, and enhances the model's generalization capability. These advantages not only contribute to improved segmentation accuracy but also lay a solid foundation for further research and applications in medical image analysis.

In our future work, we are committed to deeply optimizing and comprehensively upgrading the innovative ResUNet medical segmentation model, aiming to achieve breakthrough progress in the field of medical image analysis. This process will revolve around several core directions, with the goal of significantly enhancing the model's efficiency, accuracy, and generalization ability.

Firstly, we will focus on the fine-tuning and parameter optimization of the segmentation strategy. This includes revisiting and redesigning the various stages within the segmentation process, particularly the interface between the main body segmentation stage (without skip connections) and the detail segmentation stage (with skip connections). We will experiment with different segmentation sequences to explore which arrangement can more effectively facilitate the model's coarse-to-fine learning process, ensuring that it captures both the overall structure and the intricate

anatomical features with precision. Furthermore, for the parameter configuration of each stage, we will employ advanced hyperparameter search techniques, such as Bayesian optimization or genetic algorithms, to automatically discover the optimal settings that maximize the model's performance.

To further bolster the model's ability to recognize complex structures and fine-grained features, we plan to introduce attention mechanisms as a key upgrade. Attention mechanisms mimic the way the human visual system allocates attention, enabling the model to automatically focus on critical regions while ignoring irrelevant information when processing images. We will explore the integration of various types of attention modules (e.g., spatial attention, channel attention, or mixed attention) into the ResUNet architecture to assess their contributions to improving segmentation accuracy.

Recognizing the significance of multimodal data in medical image analysis, we also plan to expand the model's data input capabilities to incorporate information from different imaging modalities (e.g., CT, MRI, PET). This will necessitate designing multimodal fusion strategies within the model architecture, such as early fusion, mid-fusion, or late fusion, to fully leverage the complementarity between different modalities and enhance the model's robustness and accuracy.

To validate the effectiveness of these improvements, we will conduct a series of rigorous experiments, including extensive testing on diverse real-world clinical datasets. By comparing our method with the current state-of-the-art in medical image segmentation, we will comprehensively assess the impact of the new training strategy, the introduction of attention mechanisms, and the integration of multimodal data on model performance. Additionally, we will invite medical experts to participate in the evaluation process to ensure that our research findings are not only technically advanced but also highly reliable and practical in real-world applications.

Ultimately, we aspire to revolutionize the field of medical image segmentation through these endeavors, not only improving the precision and efficiency of segmentation but also promoting the widespread adoption of this technology in clinical decision support, disease diagnosis, and treatment planning. We aim to contribute significantly to medical research and clinical practice by laying a solid foundation for future research and applications in the field of medical image segmentation.

**Acknowledgements** The authors of this paper declare that the segmentation method they implemented for participation in the FLARE 2024 challenge has not used any pre-trained models nor additional datasets other than those provided by the organizers. The proposed solution is fully automatic without any manual intervention.

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
