# OpenReview forum: "Method for MICCAI FLARE24 Challenge"
_MICCAI.org/2024/Challenge/FLARE — Submitted to FLARE 2024_

### Official Review · Reviewer_DtZu · 2025-01-26
**Review of ”Method for MICCAI FLARE24 Challenge“**

**Rating:** 5
**Confidence:** 4

**Review:**

This paper presents a novel training strategy based on ResUNet to improve the accuracy of tumor segmentation in pan-cancer CT scan images. The authors optimize the model performance through phased training (removing jump connections for initial segmentation and then reintroducing jump connections for detailed segmentation). However, there is still room for improvement in DSC scores compared to nnU-Net, especially when dealing with complex structures and fine features.
Second, the title of the article may need to be more explicit about a specific issue, the resulting chart is slightly lacking, and the format of the article may need to be slightly adjusted.Also,The length of the article needs to be adjusted slightly.

---

### Official Review · Reviewer_bgz4 · 2025-01-28
**For the task of pan-cancer segmentation, the authors have modified ResUNet, but its accuracy remains relatively low. Regarding the content of the paper, I offer the following suggestions:**

**Rating:** 4
**Confidence:** 3

**Review:**

1、The title of the paper is not well-justified.
2、The qualitative and quantitative analyses are insufficiently comprehensive.

---

### Official Review · Reviewer_xLRa · 2025-03-02
**Typos and style**

**Rating:** 6
**Confidence:** 5

**Review:**

Typos and style: There are many typos or formatting issues, such as
"Abstract.. In recent years," Incorrect use of punctuation marks
“Keywords: Cancer·Segmentation·” Incorrect use of punctuation marks
“improving patient care outcomes.The complexity...” some sentences lack a space after punctuation marks.
"512x512" change to "512 x 512". x: $\times$
"3.2 evaluation measures" change to "3.2 Evaluation measures".

Please adjust the line width of Table 2.

Please change fig 2 to a meaningful comparison and adjust the CT windows.

Please modify the title to convey the core content.

---

### Decision · Program_Chairs · 2025-03-20

**Decision:**

Reject

**Comment:**

Please carefully address the reviewers' comments in the revision.